# Systematic Review of Preoperative Prognostic Biomarkers in Perihilar Cholangiocarcinoma

**DOI:** 10.3390/cancers16040698

**Published:** 2024-02-07

**Authors:** Rishaan Pawaskar, Kevin Zhang Huang, Helen Pham, Adnan Nagrial, Mark Wong, Siobhan O’Neill, Henry Pleass, Lawrence Yuen, Vincent W. T. Lam, Arthur Richardson, Tony Pang, Christopher B. Nahm

**Affiliations:** 1Department of Upper GI Surgery, Westmead Hospital, Sydney, NSW 2145, Australia; rishaan.pawaskar@health.nsw.gov.au (R.P.); helen.pham@health.nsw.gov.au (H.P.); henry.pleass@sydney.edu.au (H.P.); lawrence.yuen@health.nsw.gov.au (L.Y.); vincent.lam@health.nsw.gov.au (V.W.T.L.); arthur.richardson@health.nsw.gov.au (A.R.); tony.pang@health.nsw.gov.au (T.P.); 2Westmead Hospital, Sydney, NSW 2145, Australia; khua0227@uni.sydney.edu.au; 3Surgical Innovations Unit, Westmead Hospital, Sydney, NSW 2145, Australia; 4Westmead Clinical School, Faculty of Medicine and Health Sciences, The University of Sydney, Sydney, NSW 2006, Australia; adnan.nagrial@health.nsw.gov.au (A.N.); mark.wong@health.nsw.gov.au (M.W.); 5Crown Princess Mary Cancer Centre, Westmead Hospital, Sydney, NSW 2145, Australia; siobhan.oneill@health.nsw.gov.au; 6Macquarie University Medical School, Macquarie University, Sydney, NSW 2145, Australia

**Keywords:** perihilar cholangiocarcinoma, biomarkers, prognosis, overall survival, disease-free survival

## Abstract

**Simple Summary:**

Perihilar cholangiocarcinoma (pCCA) is an uncommon biliary tree malignancy with overall poor prognosis. Surgery is the main curative treatment; however, it carries high perioperative morbidity and mortality, with a 20–50% 5-year overall survival rate. Established prognostic factors like regional lymphatic involvement are available postoperatively; thus, invasive surgery may produce minimal oncological benefits. This systematic review aimed to summarise preoperative serum or tumour-based prognostic biomarkers in pCCA patients receiving curative intent surgery. Serum CA19-9, bilirubin, albumin, CEA, neutrophil-to-lymphocyte ratio, and platelet-to-lymphocyte ratio were most frequently associated with prognosis. Tumour matrix metalloproteinase-9 and several other molecular biomarkers were promising prognostic indicators; however, validation across multiple studies with preoperative biopsy specimens is required before their routine use. Molecular biomarkers have provided valuable insights regarding tumour biology and prognosis, thus improving staging and treatment planning in various cancers. Identification of prognostic biomarkers in pCCA may similarly improve our understanding of pCCA, potentially improving therapy and patient outcomes.

**Abstract:**

Perihilar cholangiocarcinoma (pCCA) is an uncommon malignancy with generally poor prognosis. Surgery is the primary curative treatment; however, the perioperative mortality and morbidity rates are high, with a low 5-year survival rate. Use of preoperative prognostic biomarkers to predict survival outcomes after surgery for pCCA are not well-established currently. This systematic review aimed to identify and summarise preoperative biomarkers associated with survival in pCCA, thereby potentially improving treatment decision-making. The Embase, Medline, and Cochrane databases were searched, and a systematic review was performed using the PRISMA guidelines. English-language studies examining the association between serum and/or tissue-derived biomarkers in pCCA and overall and/or disease-free survival were included. Our systematic review identified 64 biomarkers across 48 relevant studies. Raised serum CA19-9, bilirubin, CEA, neutrophil-to-lymphocyte ratio (NLR), platelet-to-lymphocyte ratio (PLR) and tumour MMP9, and low serum albumin were most associated with poorer survival; however, the cutoff values used widely varied. Several promising molecular markers with prognostic significance were also identified, including tumour HMGA2, MUC5AC/6, IDH1, PIWIL2, and DNA index. In conclusion, several biomarkers have been identified in serum and tumour specimens that prognosticate overall and disease-free survival after pCCA resection. These, however, require external validation in large cohort studies and/or in preoperatively obtained specimens, especially tissue biopsy, to recommend their use.

## 1. Introduction

Cholangiocarcinoma (CCA) is a malignant neoplasm of biliary epithelial cells and accounts for 3% of gastrointestinal cancers [1]. Perihilar cholangiocarcinoma (pCCA) is the most common subtype of CCA, arising at the bifurcation of the bile duct at the liver hilum and accounting for 50–60% of cholangiocarcinoma cases [1]. Surgical resection is the only curative treatment modality; however, only 20% of patients present with resectable disease [2]. After surgery with negative resection margins, the 5-year overall survival (OS) is 20–50% and the disease-free survival (DFS) is 30% [2,3]. It is well-understood that overall survival after pCCA resection is strongly associated with tumour T stage, positive resection margins, and regional lymph node involvement [1,2]. These survival indicators, however, are only available postoperatively as they are assessed on the resected pathology specimen.

Currently, preoperative staging for pCCA occurs mainly via imaging studies such as computed tomography (CT) and magnetic resonance imaging (MRI) [1]. Such imaging modalities only provide a single snapshot in the progression of pCCA and are limited to identifying macroscopic prognostic factors, such as radiologically evident metastatic disease and regional vascular involvement [2,3]. They do not address aspects of tumour biology such as aggressiveness, lymphovascular and perineural invasion, and micrometastatic disease, which also carry great prognostic significance [4]. Prognostic biomarkers in pCCA provide insight into tumour biology and may identify patients who are unlikely to derive oncological benefit from operative resection despite having radiologically resectable disease.

At present, serum carbohydrate antigen 19-9 (CA19-9) is the most commonly used biomarker in pCCA, used mainly to assess disease progression in the setting of systemic therapy [5]. Its prognostic significance, however, is not clear, and, importantly, levels of CA19-9 are confounded by bile duct obstruction or other hepatobiliary pathologies [6]. With improvements in understanding of cancer biology, biomarkers have been identified in various cancers to aid diagnosis, prognostication, and development of treatment targets. Various studies have evaluated a range of prognostic biomarkers for CCA in tissue and serum; however, the results are inconsistent and/or not specific for pCCA [1,3,7,8]. Biomarker assessment would likely provide valuable insights for all patients with pCCA; however, for patients planned for curative intent surgery, identification of prognostic biomarkers and their incorporation into preoperative staging for pCCA may improve patient selection for major liver resection, particularly for patients with borderline fitness for surgery. This systematic review therefore aims to identify and summarise biomarkers in blood, body fluids, and tumour tissue that prognosticate overall survival (OS) and/or recurrence-free survival (RFS) in patients who received curative intent surgery for pCCA.

## 2. Methods

### 2.1. Search Strategy

This review was performed in accordance with the PRISMA (preferred reporting items for systematic reviews and meta-analyses) guidelines and has not been registered.

Embase, Medline, PubMed, and Cochrane databases were searched to identify original articles examining the association between biomarkers and OS and/or RFS in patients who received curative intent surgery for pCCA. Biomarkers in tumour tissue and all body fluids identified by any method were included. A combination of keywords and MeSH terms for “cholangiocarcinoma”, “perihilar”, “prognosis”, “biomarkers”, and “surgery” were used (see Appendix A for full list of search terms). The searches were completed in March 2022 in all databases, and the PRISMA statement was used to screen and review results (Figure 1).

### 2.2. Study Selection

All publications were screened by title and abstract, followed by screening based on full texts by two independent authors (K.Z.H. and R.P.) according to the study’s inclusion criteria. The authors also reviewed the reference lists of included papers to manually identify any studies missed through the search strategy.

Inclusion criteria for individual studies were the following: patients treated with curative intent surgical resection for pCCA; preoperative body fluid biomarkers or tumour-based biomarkers were assessed; association between biomarkers and OS and/or RFS was examined specifically in pCCA; and English language original studies.

There was no date restriction applied. Studies were excluded if they did not specifically address prognosis and biomarkers in pCCA, did not address curative intent resection, focused on liver transplantation, were animal studies, examined radiological markers alone, or were conference abstracts. A final list of included papers was compiled, and abstracts of listed papers were reviewed by both authors to reach a consensus about their inclusion in the study. Any disagreements were resolved by discussion and/or involvement of the third author (C.N.).

### 2.3. Data Extraction

Data were extracted from studies into a proforma database listing: authors, study PMID, title, study design, total number of participants, number of participants receiving curative intent resection, biomarkers assessed, location of biomarker expression, detection method, cutoff values for delineating between positive and negative expression, number of pCCA patients positive for each biomarker, and the median and 5-year OS and DFS for all participants and those with positive biomarkers.

The quality of each included study was evaluated using the REMARK guidelines, a series of established recommendations for the reporting of research on prognostic biomarkers for cancer [9]. The guideline points were condensed into six criteria (Appendix B), adapted from the modified REMARK criteria used by Almangush et al. [10]. Each publication was assessed as to whether they fulfilled each of these six criteria.

## 3. Results

### 3.1. Study Selection

The initial literature search identified 6422 studies, as shown in the PRISMA flowchart (Figure 1). A total of 48 retrospective cohort studies were included in this review after detailed assessment of the inclusion criteria and removal of duplicates (Figure 1).

### 3.2. Adherence to REMARK Guidelines

From forty-eight included studies, eleven studies scored 6/6 on the modified REMARK guidelines, thirty-two studies scored 5/6, four studies scored 4/6, and one study scored 3/6 (Table 1). Most studies lacked clinical data reporting, particularly the timing of measurement of serum-based biomarkers relative to any preoperative interventions.

### 3.3. Biomarkers Evaluated

Across the forty-eight included studies, 64 biomarkers were evaluated, including serum proteins, blood cell counts, and molecular markers in resected pCCA specimens (Table 1). Of these, 14 biomarkers were examined by at least two different studies, while the remaining biomarkers were evaluated in single studies. Upon univariable analysis, thirty-seven biomarkers were statistically associated with OS and/or RFS, seven of which were prognostically significant in at least two studies. These were serum carbohydrate antigen 19-9 (CA19-9), serum albumin, serum bilirubin, serum carcinoembryonic antigen (CEA), neutrophil-to-lymphocyte ratio (NLR), platelet-to-lymphocyte ratio (PLR), and tumour matrix metalloproteinase (MMP9) (Table 2). Variability in cutoff values used and data reporting for each biomarker across the studies precluded meaningful meta-analysis of the data. Most tumour-derived biomarkers were evaluated by single studies.

### 3.4. Serum CA19-9

Raised preoperative CA19-9 was significantly associated with lower OS in ten out of thirteen studies [15,16,17,21,24,44,56,57,58] upon univariable analysis, and in five of these studies upon multivariable analysis [15,16,17,44,56]. There was significant heterogeneity in the cutoff values used, with twelve different values used across thirteen studies. Among the seven studies using CA19-9 cutoff values ≥100 U/mL, five studies found no statistically significant association with OS upon univariable and/or multivariable analysis, thus suggesting a possible influence of cutoff value used on the prognostic significance of this biomarker.

Four studies demonstrated significantly poorer DFS with raised CA19-9 upon univariable analysis [21,24,56,58], and one study demonstrated this upon multivariable analysis [58]. This study specifically found that CA19-9 >1000 U/mL—a much higher cutoff than the other studies—was associated with high risk of recurrence within 12 months, defined as early recurrence [58].

### 3.5. Serum Bilirubin

Raised preoperative serum bilirubin was significantly associated with OS upon univariable analysis in four [20,21,46,51] out of eight studies and remained significant in three of these upon multivariable analysis [20,46,51] (Table 2). The remaining four studies found no significant association between serum bilirubin and OS. A recent large study found that raised preoperative serum bilirubin was associated with poorer median OS of 12 months versus 15.3 months, with HR = 1.63 (95% CI 1.02–2.59, *p* = 0.040) upon multivariable analysis [51]. This study used a lower cutoff value than a majority of the identified studies and did not define time of serum bilirubin measurement in relation to any interventions, although most of their cohort did not undergo preoperative biliary drainage. An older study with a bilirubin cutoff value of 10 mg/dL had similar findings [46], while more recent studies with larger sample sizes found no statistically significant association between OS and serum bilirubin [16,21,32,42]. Differences in preoperative interventions, degree of data reporting, and exact study designs may account for the variability in results. Importantly, seven out of the eight studies did not report whether serum bilirubin was measured in the context of an obstructed biliary system or following adequate preoperative biliary drainage [16,20,21,32,42,51,58]. Two studies investigated RFS, and both found no association between serum bilirubin and RFS [21,58].

### 3.6. Serum Albumin

Seven studies examined low serum albumin and OS [11,21,31,42,44,46,51], and six of these used cutoff values between 3 and 4 mg/dL. Low serum albumin was significantly associated with poorer OS upon univariable analysis in four studies [21,31,44,51] and upon multivariable analysis in two of these studies [44,51]. One out of the seven identified studies fulfilled all the REMARK criteria and found significantly improved median OS with serum albumin > 3.5 g/dL upon univariable and multivariable analyses (HR = 0.440, 95% CI 0.22–0.88, *p* = 0.020) [44]. One study also demonstrated that albumin < 4 mg/dL was associated with poorer median RFS (10 months vs. 38 months, *p* < 0.001) upon univariable analysis; however, this was not significant upon multivariable analysis [21].

### 3.7. Serum CEA

Serum CEA was evaluated in seven studies [30,42,44,51,56,57,58], four of which reported no significant association with OS [42,51,56,57]. Two studies found that raised CEA was associated with reduced OS upon univariable and multivariable analyses [30,44]. Both studies used higher CEA cutoff value than the other studies, and most of the participants in one of these studies received neoadjuvant chemotherapy [30]. One study investigated correlation with RFS and found no significant association with CEA > 5 ng/mL [56]. Another study investigated early recurrence alone and found that CEA > 3 ng/mL was associated with early (<12 months) recurrence univariable analysis only [58].

### 3.8. Neutrophil-to-Lymphocyte Ratio (NLR)

NLR was evaluated by five studies [15,21,37,44,56]. Raised serum NLR was significantly associated with poorer OS and DFS upon univariable and multivariable analyses in two studies using similar cutoff values [21,56]. Two studies with larger cohort sizes, however, found no significant association between preoperative serum NLR and OS following R0 resection [37,44].

### 3.9. Platelet-to-Lymphocyte Ratio (PLR)

PLR was investigated by three studies [21,37,44], and raised levels were significantly associated with lower OS in two of these studies upon univariable analysis [21,44]. Upon multivariable analysis, raised serum PLR was significantly associated with poorer OS (HR = 2.207, 95% CI 1.200–4.060, *p* = 0.011) in one study [44]. One study also found that raised PLR was associated with lower RFS upon univariable but not multivariable analysis [21].

### 3.10. Tumour Matrix Metalloproteinase-9 (MMP9)

MMP9 is a tumour-based biomarker examined by two studies [47,48]. Tumour cell staining > 50% for MMP9 upon immunohistochemistry (IHC) was significantly associated with reduced OS upon multivariable analysis in both studies.

Additionally, another study assessed serum MMP9 and found that levels above a cutoff value of 201.93 ng/mL were associated with poorer OS upon multivariable analysis [32].

### 3.11. Molecular Biomarkers Evaluated by Single Studies

A total of 30 serum- and/or tissue-based molecular biomarkers were examined and found prognostically significant by single studies (Table 3). From these, molecular biomarkers demonstrating the greatest prognostic capacity, with reported hazard ratio >5 or <1, will be discussed further. These biomarkers include HMGA2, IDH1, and MUC5AC/6, as well as PIWIL2, RPL34, and DNA index.

### 3.12. High Mobility Group AT-Hook 2 (HMGA2)

The HMGA2 protein is part of a family of transcription factors modulating gene expression, replication, and DNA repair [59]. It binds to adenine–thymine regions of DNA to alter DNA structure and leads to recruitment of other protein complexes to regulate gene transcription [52,59]. HMGA2 is important during embryogenesis, but levels are largely undetectable in adult tissue [59]. Overexpression of HMGA2 is associated with numerous cancers, possibly by promoting epithelial-to-mesenchymal transition, which in turn is associated with tumour dissemination [59].

A study of 41 patients receiving curative intent surgery for pCCA found that HMGA2-positive nuclei (i.e., ≥20% HMGA2 staining) in resected tumour specimens was significantly associated with poorer 5-year OS of 32.5% versus 62.5% upon univariable analysis [52]. This association remained significant upon multivariable analysis with HR = 5.23 (95% CI 1.77–15.50, *p* = 0.003). HMGA2-positive tumours were also associated with lower DFS upon univariable and multivariable analyses in this study (HR = 3.246, 95% CI 1.296–8.133, *p* = 0.012).

### 3.13. Mucin 5AC and 6 (MUC5AC and MUC6)

MUC5AC and MUC6 are glycoproteins involved in formation of mucin gel covering and protecting the surface of epithelial cells from injury [60]. Abnormal mucin glycoproteins have been implicated in various cancers, with decreased MUC5AC expression in gastric being associated with malignant transformation [60]. Expression of such mucins may thus be considered as a potential marker of differentiation in gastrointestinal tumours [60].

A 2021 study of 30 pCCA patients who received curative intent surgery examined the association between mucin staining in tumour specimens and prognosis [26]. The PCCA specimens were divided into either low MUC5AC and six or high MUC5AC and six coexpression subgroups. This was based on the authors’ earlier work demonstrating that MUC5AC and MUC6 coexpression distinguished pCCA from distal CCA. Upon both univariable and multivariable analyses of prognostic factors, the low-expression subgroup was associated with poorer OS, with multivariable HR = 7.82 (95% CI 1.29–66.97, *p* = 0.024). Apart from higher median tumour size in the low-expression subgroup, there were no other significant differences between the two subgroups.

### 3.14. Isocitrate Dehydrogenase 1 (IDH1)

IDH1 gene mutations are implicated in various malignancies, including gliomas, leukaemia, and cholangiocarcinoma [61]. The IDH1 gene encodes the enzyme IDH1, which catalyses conversion of NADP+ to NADPH and production of a-ketoglutarate (a-KG) [61]. A-KG is an intermediary in the regulation of numerous metabolic pathways and DNA demethylase activity [61]. IDH1 mutations lead to production of 2-hydroxyglutarate instead, a competitive antagonist of a-KG, thereby contributing to increased DNA methylation and altered epigenetic control of stem and progenitor cell differentiation [61].

One 2016 study of 56 pCCA specimens after radical resection examined various cancer-associated genes and their prognostic value [43]. Interestingly, this study subdivided pCCA into extrahepatic and intrahepatic types, defined as tumours at the biliary confluence without significant hepatic extension and tumours centred at the biliary confluence extending into the intrahepatic ducts, respectively. The study did not examine the association between IDH1 mutation and OS for their entire cohort of pCCA patients. Upon univariable analysis, IDH1 mutation was associated with poorer median OS of 9.1 months versus 29.6 months (*p* = 0.043) for extrahepatic pCCA but not intrahepatic pCCA. Upon multivariable analysis of extrahepatic pCCA, IDH1 mutation was associated with poorer OS (HR = 17.844, 95% CI 3.947–17.937, *p* = 0.004) after adjusting for margin status and other clinical and pathological features.

### 3.15. PIWI-like-Protein 2 (PIWIL2)

PIWIL2 is a protein involved in stem cell self-renewal, RNA interference, chromatin remodelling, and protein translation. High expression has been demonstrated in breast, gastric, and colorectal cancer stem cells [62]. One study examined the association between prognosis and PIWIL2 expression in pCCA tissue and cell lines [19]. Among 41 patients who received curative intent surgery for pCCA, 80% had high PIWIL2 expression (overall staining and positivity score > 4), which was significantly greater than in normal biliary tract control tissue. Low PIWIL2 expression was significantly associated with improved OS (HR = 0.253, 95% CI 0.091–0.902, *p* = 0.026) and DFS (HR = 0.247, 95% CI 0.088–0.916, *p* = 0.024) upon multivariable analysis.

### 3.16. DNA Index

DNA index is another tissue-derived biomarker associated with prognosis in various solid organ malignancies and acute leukaemia [63]. It refers to the DNA content within cancer cells and may represent the genomic instability that marks cancer cells [63]. One 2015 study evaluated the prognostic value of DNA index in resected pCCA specimens of 154 patients who underwent curative intent surgery [28]. DNA index was calculated as the mean nuclear DNA content in the G0/G1 compartment of the cancer cell divided by the mean DNA content of the G0/G1 compartment of a similarly processed known diploid reference cell. Low DNA index (<1.5) was associated with longer median OS of 46.6 months versus 21.9 months (*p* = 0.003) upon univariable analysis. Upon multivariable analysis, low DNA index was associated with improved survival (HR = 0.611, 95% CI 0.402–0.929, *p* = 0.021).

### 3.17. Ribosomal Protein L34 (RPL34)

RPL34 is one of the proteins of the large 60S ribosomal subunit and facilitates rRNA stabilisation and folding, and also plays a role in cell proliferation, cell cycle regulation, and repair [64]. One study of 121 pCCA patients found that an RPL34 expression score ≥ 75 was associated with shorter OS of 1.7 years versus 3.63 years for low RPL34 expression (*p* < 0.001) upon univariable analysis [41]. Upon multivariable analysis, low RPL34 was associated with longer OS compared to a high RPL34 expression score (HR = 0.207, 95% CI 0.096–0.445, *p* < 0.001). This study also demonstrated an association between raised RPL34 expression and shorter time to recurrence (1.46 years vs. 3.73 years, *p* < 0.001) upon univariable and multivariable analyses.

## 4. Discussion

In this systematic review, serum CA19-9, bilirubin, albumin, CEA, NLR, PLR, and tumour MMP9 were the only biomarkers associated with prognosis for pCCA resected with curative intent in at least two separate studies. Several promising molecular biomarkers demonstrating prognostic significance were also identified, including HMGA2, MUC5AC, and MUC6, IDH1, PIWIL2, DNA index, and RPL34.

CA19-9 was the most frequently evaluated biomarker for pCCA, having been examined by thirteen different studies, and significantly associated with OS in ten studies [15,16,17,21,24,44,56,57,58] upon univariable analysis and five of these studies [15,16,17,44,56] upon multivariable analysis. CA19-9 is a large tetra-saccharide molecule that attaches to O-glycan moieties on cell surfaces [65], and its main clinical use is in monitoring progression of pancreaticobiliary malignancies in response to treatment [65]. High CA19-9 levels are associated with high tumour burden [66] and poorer surgical resectability [66,67], both of which are independent prognostic factors for pCCA [68]. CA19-9 is elevated with benign biliary tract obstruction too; therefore, distinguishing between tumour burden versus obstruction-related CA19-9 elevation may be difficult [69]. Further, from the thirteen studies examining CA19-9, only one study [15] clearly indicated that CA19-9 levels were measured after adequate biliary drainage, while the remaining studies did not routinely perform biliary drainage and/or did not mention the timing of the CA19-9 levels relative to biliary drainage. This is problematic given that painless obstructive jaundice is common in pCCA, and cholangitis has been associated with perioperative mortality and prognosis in some studies [2,21]. Further, 5–10% of the general population does not express the Lewis blood antigen needed to produce CA19-9, thereby leading to falsely normal values [69]. Although several studies found significant associations between raised preoperative CA19-9 and low OS, most did not report timing of measurement, inadequately assessed confounders, and some were affected by follow-up losses. The CA19-9 cutoff values also ranged from 37 to 1000 U/mL, so further studies defining the appropriate cutoff value and time of measurement are first required to be able to draw firm conclusions regarding its prognostic value.

In three out of nine studies [20,46,51], raised serum bilirubin, and, in two out of seven studies [44,51], low serum albumin were significantly associated with OS in pCCA. The studies, however, are all retrospective, with data derived from standard preoperative blood tests, and again most studies did not indicate timing of measurement and any preoperative biliary drainage in the setting of biliary tract obstruction. Ideally, such biomarker measurements should be performed in non-jaundiced patients and/or after biliary drainage to improve assessment of association between prognosis and preoperative bilirubin. The caveat, however, is that preoperative biliary drainage may not be pursued in some centres and awaiting bilirubin normalisation prior to surgery may not be realistic and instead hinder timely cancer surgery. Further, no study specifically evaluated patients with primary sclerosing cholangitis, a known risk factor for pCCA, and associated with deranged bilirubin and liver enzymes. Additionally, while albumin and bilirubin are surrogate markers of liver function, they are not distinctly associated with any known oncogenic pathway and thus do not mechanistically explain the relationship between their preoperative levels and cancer prognosis.

CEA is a less specific tumour marker for cholangiocarcinoma but is also less affected by biliary obstruction [44,68]. Among the seven included studies, two studies [30,44] found significantly reduced OS with raised CEA level at the time of diagnosis. These two studies also used higher cutoff values for CEA (7 ng/mL [44] and 8.5 ng/mL [30]) compared to the other studies, which used cutoffs of ≤5 ng/mL. This suggests that a higher CEA cutoff may be required in pCCA to determine prognosis, but this requires further evaluation with a stricter study design.

Elevated NLR and PLR have been associated with poorer survival and RFS in various malignancies, including breast, colorectal, gastric, and pancreatic cancer [70,71]. This may be related to tumour-associated inflammatory response with local neutrophil infiltration and subsequent secretion of cancer-growth-promoting cytokines such as interleukin-2,-6,-10, and VEGF [70]. Platelet infiltration is similarly associated with increased angiogenesis via VEGF, which promotes tumour growth [71]. NLR and PLR levels, however, may also be affected by various factors, including concurrent systemic infections. Only two of the included studies examining NLR and PLR clearly excluded confounders such as patients with pre-existing haematological conditions or receiving neoadjuvant chemotherapy [44,56], while one measured NLR after resolution of cholangitis [44]. Wang et al. reported a significant association between raised NLR and lower OS, while the other study found no association [44]. This may be related to the higher NLR cutoff used in the former study. PLR > 150 was significantly associated with poorer OS in one study, albeit with a wide 95% confidence interval, upon multivariable analysis. The limited number of studies investigating the prognostic role of NLR and PLR and the bias risk resulting from a retrospective design makes it difficult to conclusively recommend their use as prognostic biomarkers. Future studies also need to standardise the timepoint at which blood samples are taken for NLR and PLR analysis, ideally after resolution of confounding inflammatory/infective conditions.

MMP9 is a member of the matrix metalloproteinase family, which is involved in extracellular matrix breakdown and has been linked to the invasiveness of cancers [1]. Both studies that examined tumour MMP9 demonstrated a lower OS with >50% tumour cell staining for MMP9 [47,48]. These analyses, however, were conducted retrospectively in resected tissue samples, and, therefore, their ability to be used preoperatively (e.g., from biopsy or cytology samples) remains to be demonstrated. Another study found that raised preoperative serum MMP9 levels were significantly associated with shorter OS [32]. Such serum measurements may be more easily obtained than tumour-based measurements preoperatively; however, more studies validating their prognostic significance are required.

Several other promising molecular biomarkers associated with prognosis were identified in single studies. High tumour staining for each of HMGA2, MUC5AC/6, IDH1, PIWIL2, DNA index, and RPL34 was significantly associated with poorer OS upon multivariable analysis [19,26,28,41,43,52]. Although the identified tumour-derived biomarkers are implicated in various oncogenic pathways, their prognostic significance has not been validated across multiple studies. Studies evaluating the association between survival outcomes and biomarkers on preoperative biopsies of suspected pCCA are required to better assess their utility as prognostic biomarkers, particularly as sampling error from biopsies may further impact results. The capacity of a panel of these biomarkers to prognostically stratify patients with resectable pCCA should be assessed as this may be more informative than evaluating the biomarkers individually [72]. Patients with positive expression of multiple prognostic biomarkers may represent a cohort of patients likely to harbour micrometastatic disease and other negative pathologic factors, including perineural/lymphovascular invasion, lymph node metastases, and high tumour grade. Preoperative clinical use of validated biomarker panels would facilitate a more informed consultation with the patient regarding the perceived survival benefits of major surgical resection. This would be particularly useful in patients with borderline fitness for surgery.

The need for improving the current methods of preoperative staging is highlighted by the fact that 40% of patients who undergo potentially curative pCCA resection suffer early disease recurrence and do not survive even two years after resection [73]. Such patients likely have a high burden of micrometastatic disease and aggressive tumour biology [74]. Prognostic biomarker discovery is therefore of great importance in better staging and selecting patients for major surgical resection for pCCA.

The heterogeneity of CCA as a disease entity has been previously demonstrated in multi-institutional genomic and transcriptomic sequencing studies, with certain subtypes of CCA demonstrating poor prognosis [75]. As it remains financially burdensome and logistically unfeasible to genomically and/or transcriptomically sequence tumour tissue for all patients [76], it is important to continue the discovery and validation of biomarkers using more economical means, such as immunohistochemistry and/or enzyme-linked immunoassays. The identification of reliable biomarkers may lead to a more nuanced selection of treatment strategies for patients with aggressive disease. Indeed, this has long been the norm in breast cancer, where oestrogen, progesterone, and HER2 receptor expression are highly influential in determining treatment in neoadjuvant, adjuvant, and metastatic settings [77]. There is some retrospective evidence demonstrating that a similar approach using other biomarkers may be useful in determining a neoadjuvant versus surgery-first approach in patients with resectable pancreas cancer [78]. Identification of prognostic biomarkers in pCCA may similarly enhance staging and guide treatment for patients with pCCA.

The limitations of this systematic review are predominantly related to the retrospective nature of the included studies, significant variability in study design, prognostic factor measurement, and outcome reporting across included studies. Importantly, most studies did not clearly define time of biomarker measurement relative to any preoperative interventions, and the ideal time point for measurements has not been established across the literature. These factors make it difficult to draw definite conclusions about prognostic markers. Additionally, variability in reported biomarker cutoffs across studies precluded a meaningful meta-analysis. For future studies, adherence to REMARK criteria, particularly by clearly defining time of biomarker measurements and/or relation to preoperative interventions, will improve the standardisation of data reporting and may potentially enable a meta-analysis. Pragmatically, biomarker measurements, particularly serum biomarkers, should be at diagnosis and after any biliary drainage to aid prognostication and surgical decision-making in patients otherwise appropriate for curative intent treatment. Serial biomarker measurements to observe dynamic trends may provide additional information regarding tumour biology and progression, akin to the concept of PSA velocity for prostate cancer [79]; however, this remains to be validated for biomarkers in pCCA. Population differences in biomarker profiles according to comorbidities, particularly liver disease, may also exist, thus reinforcing the need for improved assessment and standardization of cohorts in future studies. Many CCA biomarker studies were excluded from this review as more than one CCA subtype was analysed without distinguishing pCCA. pCCA is clinicopathologically and genomically distinct from other CCA subtypes [1], so identifying biomarkers specific to pCCA is important for prognostication.

## 5. Conclusions

In conclusion, this systematic review provides a comprehensive overview of the current evidence regarding serum and tumour biomarkers and prognosis in pCCA. Prospective studies on biomarkers in preoperative biopsies or blood are required for their validation and subsequent clinical utility. This may, in the future, enable a biomarker-based approach to pCCA staging, leading to improved treatment decision-making for patients with disease.

## Figures and Tables

**Figure 1 cancers-16-00698-f001:**
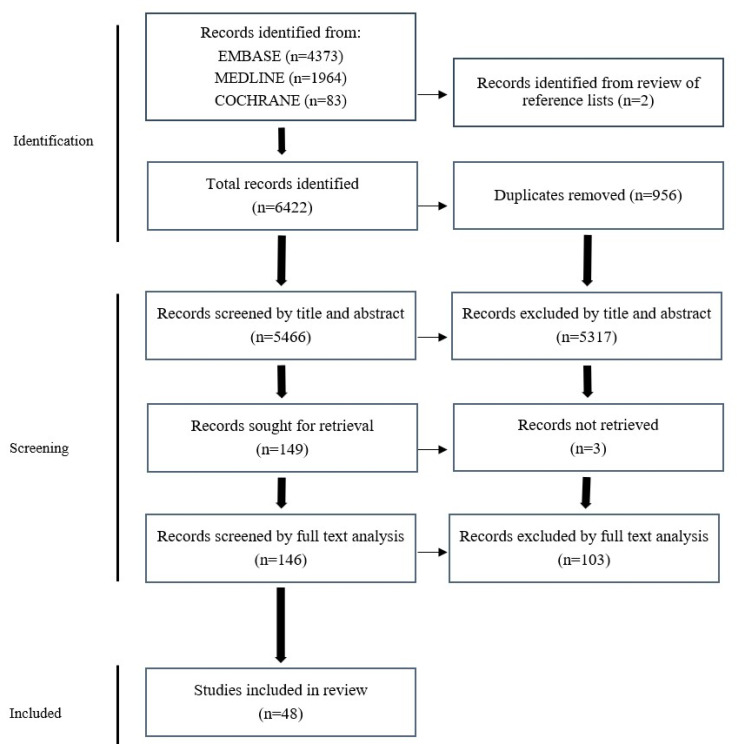
PRISMA flowchart of study identification, screening, and inclusion.

**Table 1 cancers-16-00698-t001:** All studies and biomarkers included with modified REMARK score. Prognostic significance is described as N = not significant, U = significant upon univariable analysis, or M = significant upon multivariable analysis. OS, overall survival. RFS, recurrence-free survival. IHC, immunohistochemistry. CA19-9, carbohydrate antigen 19-9. CEA, carcinoembryonic antigen. TAM, tumour-associated macrophages. TEM, Tie2-expressing monocytes. qPCR, quantitative polymerase chain reaction. MLR, monocyte-to-lymphocyte ratio. NLR, neutrophil-to-lymphocyte ratio. PIWIL2, Piwi-like RNA-mediated gene silencing 2. TLR, thrombocyte-to-lymphocyte ratio. P-4E-BP1, phosphor 4E-BP1. ARID1B, AT-Rich interaction domain 1B. RBM10, RNA binding motif protein 10. CYFRA21-1, cytokeratin 19 fragment. MUC5A, mucin 5A. MUC6, mucin 6. OLFM4, olfactomedin 4. PD-L1, programmed death ligand 1. MMP9, matrix metalloproteinase 9. TCF7, transcription factor 7. VEGF, vascular endothelial growth factor. HDGF, hepatoma-derived growth factor. NGAL, neutrophil-gelatinase-associated lipocalin. TROP2, trophoblast cell surface antigen 2. mGPS, modified Glasgow prognostic score. PLR, platelet-to-lymphocyte ratio. PNI, prognostic nutritional index. EGFR, epidermal growth factor. ERCC1, excision repair cross-complementing group 1. TP, thymidine phosphorylase. TP53, tumour protein 53. LMRc, change in lymphocyte-to-monocyte ratio. RPL34, ribosomal protein L34. ALK, anaplastic lymphoma kinase. BRAF, b-raf protein. IDH, isocitrate dehydrogenase. PIK3CA, phosphatidylinositol-4,5-bisphosphate 3-kinase, catalytic subunit alpha. PBRM1, polybromo 1. CRP, C reactive protein. Gal-3, galectin-3. ALT, alanine transaminase. AST, aspartate transaminase. IL8, interleukin 8. ANXA10, annexin A10. GGT, gamma glutamyl transferase. HMGA2, high mobility group AT-hook 2. CA125, cancer antigen 125. MSLN, mesothelin. NGF, nerve growth factor. CONUT, controlling nutritional status. ALP, alkaline phosphatase.

Authors	Biomarker	Cutoff	Detection Site	Detection Method	Total Participants with pCCA	Participants with Positive Biomarker	REMARK Score	Prognostic Significance (N/U/M)
Abdel Wahab et al., 2016 [11]	Albumin	not specified	Serum	not specified	234	not specified	3/6	N
	Bilirubin	not specified	Serum	not specified	234	not specified		N
	CA19-9	not specified	Serum	not specified	234	not specified		M (OS)
Atanasov et al., 2016 [12]	TAM	>25% cell density	Tumour	IHC	45	12	6/6	U (OS) M (RFS)
Atanasov et al., 2016 [13]	TEM	expression score 2	Tumour	IHC	45	11	6/6	M (OS)
Atanasov et al. 2018 [14]	miR-126	not specified	Tumour	qPCR	45	18	5/6	U (OS)
Bird et al., 2019 [15]	CA19-9	>46 U/ml	Serum	not specified	56	22	5/6	M (OS)
	MLR	≥3	Serum	not specified	56	31		N
	NLR	≥3	Serum	not specified	56	23		N
Cai et al., 2014 [16]	CA19-9	>150 U/ml	Serum	not specified	168	74	5/6	M (OS)
	Bilirubin	>10 mg/dL	Serum	not specified	168	96		N
Chen P et al., 2016 [17]	CA19-9	73.5–325 U/mL	Serum	not specified	235	78	5/6	M (OS)
	CA19-9	>325 U/ml	Serum	not specified	235	79		M (OS)
Chen W et al., 2015 [18]	Complete vs. partial Wnt pathway activation	Wnt2+/3+ and b-catenin + and TCF4+ vs. Wnt2+/3+ and b-catenin + or TCF4+	Tumour	IHC	129	101	4/6	M (OS)
Chen YJ et al., 2015 [19]	PIWIL2	expression score ≥ 4	Tumour	IHC	41	33	5/6	M (OS + RFS)
Cheng et al., 2007 [20]	Bilirubin	>10 mg/dL	Serum	not specified	75	not specified	5/6	M (OS)
Dumitrascu et al., 2013 [21]	Albumin	>4 g/dL	Serum	not specified	90	not specified	4/6	U (OS + RFS)
	Bilirubin	>10.4 mg/dl	Serum	not specified	90	not specified		U (OS + RFS)
	CA19-9	>200 U/mL	Serum	not specified	90	not specified		U (OS + RFS)
	Haemoglobin	>12.5 mg/dL	Serum	not specified	90	not specified		U (OS + RFS)
	Leukocyte count	>8000/mmc	Serum	not specified	90	not specified		U (OS + RFS)
	NLR	>3.3	Serum	not specified	90	not specified		U (OS) M (RFS)
	TLR	>184	Serum	not specified	90	not specified		U (OS + RFS)
	Platelets	>312,000/mmc	Serum	not specified	90	not specified		U (OS + RFS)
Fang et al., 2014 [22]	P-4E-BP1	>25% positive cells	Tumour	IHC	61	31	5/6	U (OS + RFS)
Feng et al., 2021 [23]	ARID1B	n/a	Tumour	NGS	63	58	5/6	M (RFS)
	RBM10	n/a	Tumour	NGS	63	58		N
Hu et al., 2016 [24]	CA19-9	>100 U/mL	Serum	Not specified	381	not specified	5/6	U (OS + RFS)
Huang et al., 2015 [25]	CYFRA21-1	>2.27 ng/mL	Serum	ECL	31	22	5/6	U (OS + RFS)
Ishida et al., 2019 [26]	MUC5A and MUC6	<25% expression	Tumour	IHC	30	13	6/6	M (OS)
Jun et al., 2020 [27]	OLFM4		Tumour	IHC	54	14	5/6	N
Kamphues et al., 2015 [28]	DNA index	>1.5	Tumour	Image cytometry	154	71	5/6	M (OS)
Kriegsmann et al., 2019 [29]	PD-L1	>5% positive cells	Tumour	IHC	57	3	5/6	N
Kuriyama et al., 2020 [30]	CA19-9	≥25 U/ml	Serum	not specified	49	35	6/6	N
	CEA	≥8.5 ng/mL	Serum	not specified	49	7		M (OS)
Li et al., 2020 [31]	Albumin	>35 g/L	Blood	not specified	292	not specified	5/6	U (OS)
	Mean Platelet Volume	>13	Blood	not specified	292	not specified		U (OS)
	Platelet Distribution Width	>16.55	Blood	not specified	292	121		M (OS)
	Platelets	>300 × 10^9^	Blood	not specified	292	not specified		N
Li et al., 2019 [32]	Bilirubin	>100 mg/L	Blood	not specified	181	not specified	5/6	N
	MMP9	>201.93 ng/mL	Blood	ELISA	181	not specified		M (OS)
Liu et al., 2019 [33]	TCF7	not specified. Comparison of low vs. high	Tumour	IHC	160	76	5/6	M (OS)
Liu et al., 2011 [34]	VEGF	>25% staining	Tumour	IHC	58	42	5/6	M (OS)
	HDFG	HDGF labelling index > 166.91	Tumour	IHC	58	27		U (OS)
Nair et al., 2018 [35]	NGAL	>345 Histoscore	Tumour	IHC	54	not specified	5/6	N
Ning et al., 2013 [36]	TROP2	Score > 4	Tumour	IHC	70	43	6/6	M (OS)
Okuno et al., 2016 [37]	mGPS	mGPS ≥ 1	Serum	not specified	534	112	5/6	M (OS)
	NLR	≥3	Serum	not specified	534	158		N
	PLR	≥150	Serum	not specified	534	359		N
	PNI	≥40	Serum	not specified	534	362		N
Park et al., 2015 [38]	C-met	Staining ≥ 30%	Tumour	IHC	53	19	5/6	N
	EGFR	Staining ≥ 30%	Tumour	IHC	53	not specified		N
	VEGF	Staining ≥ 30%	Tumour	IHC	53	not specified		N
Park et al., 2013 [39]	ERCC1	Staining > 10%	Tumour	IHC	41	9	5/6	N
	Survivin	Staining > 10%	Tumour	IHC	41	14		U (OS)
	TP	Staining > 10%	Tumour	IHC	41	25		N
	TP53	Staining > 10%	Tumour	IHC	41	18		N
	Cyclin D1	Staining > 10%	Tumour	IHC	41	8		N
Peng et al., 2020 [40]	LMRc	Postop LMR > preop LMR	Serum	not specified	254	125	6/6	U (OS) M (RFS)
Qian et al., 2020 [41]	RPL34	Expression ≥ 75	Tumour	IHC	121	94	5/6	M (OS + RFS)
Ramacciato et al., 2010 [42]	Albumin	Low < 3 g/dL	Serum	not specified	30	21	4/6	N
	Bilirubin	>3 mg/dL	Serum	not specified	30	11		N
	CA19-9	>400 ng/mL	Serum	not specified	30	17		N
	CEA	>2 U/L	Serum	not specified	30	18		N
Ruzzenente et al., 2016 [43]	ALK	n/a	Tumour	NGS	18	1	5/6	U (OS)
	BRAF	n/a	Tumour	NGS	18	1		N
	IDH	n/a	Tumour	NGS	18	2		M (OS)
	KRAS	n/a	Tumour	NGS	18	12		N
	PIK3CA	n/a	Tumour	NGS	18	3		N
	PBRM1	n/a	Tumour	NGS	18	3		N
	TP53	n/a	Tumour	NGS	18	11		M (OS)
Saito et al., 2016 [44]	Albumin	<3.5 mg/dL	Serum	not specified	121	26	6/6	M (OS)
	CRP	>0.5 mg/dL	Tumour	not specified	121	56		M (OS)
	CA19-9	>300 U/mL	Tumour	not specified	121	17		U (OS)
	CEA	>7 ng/mL	Tumour	not specified	121	14		M (OS)
	PLR	>150	Tumour	not specified	121	53		U (OS)
Shimura et al., 2017 [45]	Gal-3	>50% staining	Tumour	IHC	21	11	6/6	N
	Intranuclear Gal-3	>5% intranuclear staining	Tumour	IHC	21	not specified		N
Su et al., 1996 [46]	ALT	>120 U/L	Serum	not specified	44	13	5/6	N
	Albumin	<30 mg/L	Serum	not specified	44	45		N
	AST	>135 U/L	Serum	not specified	44	6		N
	Bilirubin	>10 mg/dL	Serum	not specified	44	17		M (OS)
Sun et al., 2014 [47]	MMP9	>50% staining	Tumour	IHC	58	27	6/6	M (OS)
Sun Q et al., 2015 [48]	IL8	median score—not specified	Tumour	IHC	62	35	6/6	M (OS)
	MMP9	>50% staining	Tumour	IHC	62	29		M (OS)
Sun Q et al., 2018 [49]	Pontin	>5% staining	Tumour	IHC	86	34	5/6	M (OS)
Sun R et al., 2019 [50]	ANXA10	>5% staining	Tumour	IHC	128	not specified	5/6	M (OS)
Sun Z et al., 2020 [51]	Albumin	<35 g/L	Serum	not specified	110	173	5/6	M (OS)
	ALT	≥80 U/L	Serum	not specified	110	225		N
	AST	≥70 U/L	Serum	not specified	110	202		N
	GGT	≥90 U/L	Serum	not specified	110	288		N
	Bilirubin	≥4 mg/Dl	Serum	not specified	110	242		M (OS)
	CRP	≥5 mg/L	Serum	not specified	110	189		N
	CA19-9	>0.001 U/mL	Serum	not specified	110	not specified		N
	CEA	≥5 ng/mL	Serum	not specified	110	71		N
Takahashi et al., 2021 [52]	HMGA2	>50% staining	Tumour	IHC	41	21	6/6	M (OS)
Takihata et al., 2021 [53]	CA125 and MSLN coexpression	>50% staining	Tumour	IHC	31	15	5/6	U (OS)
Thelen et al., 2008 [54]	CD31	Average count > 20	Tumour	IHC	60	38	5/6	M (OS) U (RFS)
Urabe et al., 2016 [55]	NGF	>30%	Tumour	IHC	59	35	5/6	N
Wang et al., 2020 [56]	CA19-9	>37 ng/ml	Serum	not specified	94	not specified	5/6	U (OS)
	CEA	>5 ng/mL	Serum	not specified	94	not specified		N
	NLR	>3.6	Serum	not specified	94	51		M (OS + RFS)
	PNI	>43.7	Serum	not specified	94	50		M (OS + RFS)
	CONUT Score	>3	Serum	not specified	94	31		M (OS + RFS)
Yoo et al., 2021 [57]	ALT	>40 U/L	Serum	not specified	196	not specified	5/6	N
	AST	≥40 U/L	Serum	not specified	196	not specified		N
	ALP	>115 U/L	Serum	not specified	196	not specified		N
	CA19-9	>37 U/mL	Serum	not specified	196	not specified		M (OS)
	CEA	>5 ng/ml	Serum	not specified	196	not specified		N
Zhao et al., 2020 [58]	Bilirubin	≥142.4 µmol/L	Serum	not specified	335	168	4/6	M (OS)
	CA19-9	≥1000 U/mL	Serum	not specified	335	68		U (OS) M (RFS)
	CEA	≥3.0 ng/mL	Serum	not specified	335	168		U (RFS)

**Table 2 cancers-16-00698-t002:** Biomarkers demonstrated to be prognostically significant in at least two studies. Hazard ratios reported for biomarker above cutoff value as compared with biomarker below cutoff value unless indicated. N = not significant. NR = not reported.

Author, Year	Cutoff Value	No. of pCCA Resections	Endpoint	Univariable Analysis mOS Months, + vs. − or HR	Multivariable Analysis	Unfulfilled Modified REMARK Criteria
Serum CA19-9						
Abdel Wahab et al., 2016 [11]	CA 19-9 considered as continuous variable	243	OS	Significant (numbers not reported)	HR = 1.01 (1.01–1.02) *p* = 0.000	2: Clinical data 5: Statistics 6: Classical prognostic factors
Bird et al., 2019 [15]	>46 U/mL	56	OS	41.3 vs. 66.0 mths *p* = 0.024	HR = 3.24 (1.37–7.69) *p* = 0.007	2: Clinical data
Cai et al., 2014 [16]	>150 U/mL	168	OS	22 vs. 44 mths *p* = 0.001	HR = 2.23 (1.14–4.39) *p* = 0.020	2: Clinical data
Chen et al., 2016 [17]	>73.5 but <325 U/mL	235	OS	HR = 1.63 (1.14–2.32) *p* = 0.008	(vs. <73.5 U/mL) HR = 1.70 (1.18–2.44) *p* = 0.004	2: Clinical data
	≥325 U/mL	235	OS	(vs. <73.5 U/mL) HR = 2.39 (1.68–3.41) *p* < 0.001	HR = 2.30 (1.60–3.30) *p* < 0.001	
Dumitrascu et al., 2013 [21]	200 U/mL	90	OS	13 vs. 43 mths *p* < 0.001	N	2: Clinical data 6: Classical prognostic factors
			RFS	12 vs. 35 mths *p* = 0.004	N	
Hu et al., 2016 [24]	100 U/mL	381	OS	23 vs. 39.7 mths *p* = 0.039	N	2: Clinical data
			RFS	16.7 vs. 23.6 mths *p* = 0.018	N	
Kuriyama et al., 2020 [30]	25 U/mL	49	OS	49.4 mths vs. not reached *p* = 0.355	NR	
Ramacciato et al., 2010 [42]	400 ng/mL	30	OS	16 vs. 19 mths *p* = 0.460	NR	2: Clinical data 5: Statistics
Saito et al., 2016 [44]	300 U/mL	121	OS	27.6 vs. 74.6 mths *p* = 0.003	HR = 1.00 (0.46–2.17) *p* = 0.999	
Sun Z et al., 2020 [51]	>0.001 U/L	110	OS	12.1 vs. 13 mths *p* = 0.678	NR	2: Clinical data
Wang et al., 2020 [56]	37 ng/mL	94	OS	HR = 2.02 (1.06–3.84) *p* = 0.032	HR = 0.85 (0.39–1.87) *p* = 0.692	2: Clinical data
			RFS	HR = 3.58 (1.69–7.59) *p* = 0.001	HR = 1.75 (0.76–4.05) *p* = 0.189	
Yoo et al., 2021 [57]	37 U/mL	196	OS	HR = 2.16 (1.48–3.17) *p* < 0.001	HR = 2.06 (1.37–3.10) *p* < 0.001	2: Clinical data
Zhao et al., 2020 [58]	>1000 U/mL	335	OS	25 mths vs. 33 mths *p* = 0.002	NR	2: Clinical data 5: Statistics
			RFS < 12 mths	OR = 1.662 (1.071–2.581) *p* = 0.024	OR = 2.205 (1.208–4.026) *p* = 0.010	
Serum bilirubin						
Abdel Wahab et al., 2016 [11]	NR	243	OS	N	NR	2: Clinical data 5: Statistics 6: Classical prognostic factors
Cai et al., 2014 [16]	>10 mg/dL	168	OS	39 vs. 33 mths *p* = 0.436	NR	2: Clinical data
Cheng et al., 2007 [20]	>10 mg/dL	75	OS	Significant (numbers not reported)	HR = 2.0 (1.5–2.5) *p* = 0.04	2: Clinical data
Dumitrascu et al., 2013 [21]	>10.4 mg/dL	90	OS	17 vs. 42 mths *p* = 0.010	N	2: Clinical data 6: Classical prognostic factors
			RFS	12 vs. 31 mths *p* = 0.011	N	
Li et al., 2019 [32]	>10 mg/dL	181	OS	41.6 vs. 40.7 mths *p* = 0.868	NR	2: Clinical data
Ramacciato et al., 2010 [42]	>3 mg/dL	30	OS	18 vs. 16 mths *p* = 0.259	NR	2: Clinical data 5: Statistics
Su et al., 1996 [46]	>10 mg/dL	44	OS	6 vs. 18 mths *p* = 0.006	HR = 2.44 (1.01–5.88) *p* = 0.001	2: Clinical data
Sun Z et al., 2020 [51]	>4 mg/dL	110	OS	12 vs. 15.3 mths *p* = 0.018	HR = 1.63 (1.02–2.59) *p* = 0.040	2: Clinical data
Zhao et al., 2020 [58]	>142.4 µmol/L	335	RFS <12 mths	OR = 1.207 (0.780–1.868) *p* = 0.398	NR	2: Clinical data 5: Statistics
Serum albumin *						
Abdel Wahab et al., 2016 [11]	NR	243	OS	N	NR	2: Clinical data 5: Statistics 6: Classical prognostic factors
Dumitrascu et al., 2013 [21]	<4 g/dL	90	OS	44 vs. 13 mths *p* < 0.001	N	2: Clinical data 6: Classical prognostic factors
			RFS	38 vs. 10 mths *p* < 0.001	N	
Li et al., 2020 [31]	<3.5 g/dL	292	OS	HR 0.72 (0.52–0.99) *p* = 0.041	NR	2: Clinical data
Ramacciato et al., 2010 [42]	<3 g/dL	30	OS	19 vs. 11 mths *p* = 0.691	NR	2: Clinical data 5: Statistics
Saito et al., 2016 [44]	<3.5 g/dL	121	OS	88.1 vs. 34.6 mths *p* = 0.003	HR = 0.44 (0.22–0.88) *p* = 0.020	
Su et al., 1996 [46]	<3 g/dL	44	OS	N and NR	NR	2: Clinical data
Sun Z et al., 2020 [51]	<3.5 g/dL	110	OS	19.4 vs. 9.2 mths *p* < 0.001	HR = 0.65 (0.45–0.94) *p* = 0.023	2: Clinical data
Serum CEA						
Kuriyama et al., 2020 [30]	>8.5 ng/mL	49	OS	21.2 mths vs. not reached 5 yr OS: 39.2% vs. 51.6% *p* < 0.001	HR = 10.516 (2.213–49.971) *p* = 0.003	
Ramacciato et al., 2010 [42]	>2 U/ml	30	OS	11 mths vs. 19 mths *p* = 0.465	NR	2: Clinical data 5: Statistics
Saito et al., 2016 [44]	>7 ng/mL	121	OS	20.5 mths vs. 88.1 mths *p* < 0.001	HR = 5.033 (2.273–11.14) *p* <0.001	
Sun Z et al., 2020 [51]	>5 ng/ml	110	OS	11.4 mths vs. 12.1 mths *p* = 0.635	NR	2: Clinical data
Wang et al., 2020 [56]			RFS	HR = 1.30 (0.75–2.26) *p* = 0.348		2: Clinical data
Yoo et al., 2021 [57]	>5 ng/ml	94	OS	HR = 1.29 (0.73–2.29) *p* = 0.374	NR	2: Clinical data
Zhao et al., 2020 [58]	>5 ng/mL	196	OS	HR = 1.82 (0.97–3.40) *p* = 0.062	NR	2: Clinical data
	>3 ng/mL	335	RFS < 12 mths	OR = 1.662 (1.071–2.581) *p* = 0.024	OR = 1.279 (0.775–2.108) *p* = 0.336	2: Clinical data 5: Statistics
Serum NLR						
Bird et al., 2019 [15]	>3	56	OS	33.7 vs. 53.6 mths *p* = 0.42	NR	2: Clinical data
Dumitrascu et al., 2013 [21]	>3.3	90	OS	15 vs. 43 mths *p* < 0.001	HR = 0.76 (0.57–1) *p* = 0.053	2: Clinical data
			RFS	11 vs. 38 mths *p* < 0.001	HR = 0.78 (0.62–0.98) *p* = 0.036	
Okuno et al., 2015 [37]	>5	538	OS	39 vs. 28 mths *p* = 0.477	NR	1: Cohort overview
Saito et al., 2016 [44]	>2.5	121	OS	49.3 mths vs. 74.6 mths *p* = 0.225	NR	
Wang et al., 2020 [56]	>3.6	94	OS	HR = 2.78 (1.59–4.86) *p* < 0.001	HR = 2.27 (1.22–4.24) *p* = 0.010	2: Clinical data
			RFS	HR = 2.17 (1.29–3.66) *p* = 0.004	HR = 1.83 (1.03–3.24) *p* = 0.038	
Serum PLR						
Dumitrascu et al., 2013 [21]	>184	90	OS	17 mths vs. 43 mths *p* = 0.002	NR	2: Clinical data 6: Classical prognostic factors
			RFS	12 mths vs. 35 mths *p* = 0.004	NR	
Okuno et al., 2015 [37]	<150 ≥150 ≥300	534	OS	36 mths 44.4 mths 42 mths *p* = 0.409	NR	1: Cohort overview
Saito et al., 2016 [44]	>150	121	OS	49.3 mths 5 yr OS 43.8% vs. 62.4% *p* = 0.012	HR = 2.207 (1.200–4.060) *p* = 0.011	
Serum and Tumour MMP9						
Li et al., 2019 [32]	>201.93 ng/mL (Serum)	181	OS	34.5 vs. 50.9 mths *p* < 0.001	HR = 5.19 (CI NR) *p* < 0.01	2: Clinical data
Sun et al., 2014 [47]	>50% tumour expression	58	OS	15 vs. >40 mths *p* < 0.001	HR = 4.30 (1.49–12.43) *p* = 0.007	
Sun et al., 2015 [48]	>50% tumour expression	62	OS	NR *p* < 0.022	HR = 3.27 (1.22–6.94) *p* = 0.016	

* HR compares albumin level below cutoff versus above cutoff.

**Table 3 cancers-16-00698-t003:** Biomarkers examined by a single study and statistically significant as prognostic indicator of OS in pCCA. IHC, immunohistochemistry. miR-126, micro RNA 126. TAM, tumour-associated macrophages. TEM, Tie2-expressing monocytes. PIWIL2, piwi-like RN- mediated gene silencing 2. P-4E-BP1, phosphor 4E-BP1. CYFRA21-1, cytokeratin 19 fragment. ECLiA, electrochemiluminscent immunoassay. MUC5A, mucin 5A. MUC6, mucin 6. MPV, mean platelet volume. PDW, platelet distribution width. HDGF, hepatoma-derived growth factor. VEGF, vascular endothelial growth factor. TCF7, transcription factor 7. TROP2, trophoblast cell surface antigen 2. mGPS, modified Glasgow prognostic score. LMRc, change in lymphocyte-to-monocyte ratio. RPL34, ribosomal protein L34. ALK, anaplastic lymphoma kinase. IDH, isocitrate dehydrogenase. P53, protein 53. CRP, C reactive protein. IL8, interleukin 8. ANXA10, annexin 10. PNI, prognostic nutritional index. HMGA2, high mobility group AT-hook 2. CONUT, controlling nutritional status.

Author, Year	Biomarker	Detection	Univariable Analysis	Multivariable Analysis
Atanasov et al. 2018 [14]	miR-126	Tumour IHC	mOS 8 vs. 24 mths *p* = 0.004	NR
Atanasov et al., 2016 [13]	TEM	Tumour IHC	5 yr OS 14.9% vs. 56.5% *p* = 0.026	HR 2.90 (1.01–8.36) *p* = 0.045
Atanasov et al., 2016 [12]	TAM	Tumour IHC	HR 0.32 (0.13–0.78) *p* = 0.013	HR 0.856 (0.29–2.56) *p* = 0.78
Chen et al., 2015 [19]	PIWIL2	Tumour IHC	mOS 25.3 vs. 37.7 mths *p* = 0.013	HR 0.253 (0.09–0.90) *p* = 0.026
Dumitrascu et al., 2013 [21]	Haemoglobin	Blood	mOS 17 vs. 43 mths *p* = 0.015	NR
	Leukocyte count	Blood	mOS 17 vs. 43 mths *p* = 0.010	NR
	Platelet count	Blood	mOS 14 vs. 44 mths *p* < 0.001	NR
Fang et al., 2014 [22]	P-4E-BP1	Tumour IHC	mOS 19.5 vs. 29.9 mths *p* = 0.006	NR
Huang et al., 2015 [25]	CYFRA21-1	ECLiA	5 yr OS 36.3% vs. 87.5% *p* < 0.01	NR
Ishida et al., 2019 [26]	MUC5AC and MUC6	Tumour IHC	HR 4.67 (1.42–17.98) *p* = 0.010	HR 7.82 (1.29–66.97) *p* = 0.024
Kamphues et al., 2015 [28]	DNA index	Tumour image cytometry	mOS 21.9 vs. 46.6 mths *p* = 0.010	HR 0.61 (0.40–0.93) *p* = 0.021
Li et al., 2020 [31]	MPV	Blood	HR 1.69 (1.23–2.31) *p* = 0.01	HR 1.24 (0.89–1.74) *p* = 0.206
	PDW	Blood	HR 3.12 (2.33–4.19) *p* < 0.01	HR 2.52 (1.83–3.47) *p* < 0.001
Liu et al., 2011 [34]	HDGF	Tumour IHC	Survival rate 35.7% vs. 73.3% *p* = 0.003	HR 4.36 (1.45–13.10) *p* = 0.009
	VEGF	Tumour IHC	Survival rate 45.2% vs. 81.3% *p* = 0.018	HR 3.913 (0.83–18.43) *p* = 0.084
Liu et al., 2019 [33]	TCF7	Tumour IHC	3 yr OS 59.0% vs. 23.0% *p* = 0.019	HR 2.06 (CI NR) *p* = 0.024
Ning et al., 2013 [36]	TROP2	Tumour IHC	mOS 37 vs. >70 mths *p* = 0.001	HR 3.26 (1.47–7.21) *p* = 0.004
Okuno et al., 2016 [37]	mGPS	Blood	5 yr OS 26.3% vs. 41.9% *p* < 0.001	HR 1.58 (1.21–2.06) *p* = 0.001
Peng et al., 2020 [40]	LMRc	Blood	mOS 20 vs. 36 mths *p* = 0.001	NR
Qian et al., 2020 [41]	RPL34	Tumour IHC	mOS 1.70 vs. 3.63 yrs *p* < 0.001	HR 0.25 (0.12–0.54) *p* = 0.001
Ruzzenente et al., 2016 [43]	ALK	Tumour PCR	mOS 5 vs. 34.9 mths *p* < 0.001	NR
	IDH1	Tumour IHC	mOS 9.1 vs. 29.6 mths *p* = 0.043	HR 17.84 (3.95–17.40) *p* = 0.004
	P53	Tumour PCR	mOS 15.4 vs. 32.5 mths *p* = 0.019	HR 2.706 (1.11–8.21) *p* = 0.039
Saito et al., 2016 [44]	CRP	Blood	mOS 44.6 vs. >50 mths *p* < 0.001	HR 3.29 (1.80–6.03) *p* < 0.001
Sun Q et al., 2015 [48]	IL8	Tumour IHC	mOS 17 vs. >42 mths *p* = 0.010	HR 2.46 (0.92–8.54) *p* = 0.005
Sun Q et al., 2018 [49]	Pontin	Tumour IHC	mOS 18 vs. > 40 mths *p* = 0.002	HR 2.883 (1.26–6.72) *p* = 0.001
Sun R et al., 2019 [50]	ANXA10	Tumour IHC	3 yr OS 13.1% vs. 52.0% *p* < 0.001	HR 2.25 (1.34–3.84) *p* = 0.026
Takahashi et al., 2021 [52]	HMGA2	Tumour IHC	HR 3.05 (1.19–7.80) *p* = 0.015	HR 5.23 (1.77–15.50) *p* = 0.003
Wang et al., 2020 [56]	CONUT score	Blood	HR 3.77 (2.21–6.43) *p* < 0.001	HR 4.01 (1.97–8.18) *p* < 0.001
	PNI	Blood	HR 0.36 (0.21–0.61) *p* < 0.001	HR 0.26 (0.14–0.49) *p* < 0.001

NR, not reported.

## Data Availability

No new data were created as part of this systematic review.

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
