# Peer review of "Systematic Review of Preoperative Prognostic Biomarkers in Perihilar Cholangiocarcinoma"

_cancers, 2024, doi:10.3390/cancers16040698_

Round 1

Reviewer 1 Report

Comments and Suggestions for Authors

Dear Authors

your review on the significance and use of prognostic markers in perihilar CCA is very useful to clinicians and very well described.

I have no peculiar comments and remarks to make since all the aspects in your paper have been properly addressed, including the current limitations of the most used markers.

Author Response

We thank reviewer 1 for their consideration and input on our manuscript. 

Reviewer 2 Report

Comments and Suggestions for Authors

Pawaskar et al presented a systematic review of Preoperative Prognostic Biomarkers in Perihilar Cholangiocarcinoma. The authors attempted to compile the preoperative biomarkers associated with survival in pCCA and identified 64 biomarkers across 48 relevant studies. This is a good piece of work and guides the understanding of pCCA to potentially improve the therapy and patient outcome. This review is organized well and presented lucidly. The selected topic is relevant and need of the hour. Recent literature is considered and a conclusion is drawn based on the data evaluation. Limitations of the study are also mentioned in the manuscript. Based on the merit, the manuscript can be accepted in its current format.

Author Response

We thank Reviewer 2 for their consideration and input on our manuscript. 

Reviewer 3 Report

Comments and Suggestions for Authors

Perihilar cholangiocarcinoma, a rare and potentially fatal malignancy, is primarily addressed through surgery. Despite the treatment, the overall prognosis remains unfavorable, marked by high perioperative mortality and morbidity rates and low 5-year survival rates. This systematic review aimed to identify preoperative biomarkers linked to survival in pCCA, contributing to more informed treatment decisions. Examining 48 studies, the review identified 64 biomarkers in serum and/or tissue associated with both overall and disease-free survival. Elevated levels of serum CA19-9, bilirubin, CEA, neutrophil-to-lymphocyte ratio (NLR), platelet-to-lymphocyte ratio, and tumor MMP9, alongside low serum albumin, were correlated with poorer survival. Additionally, promising molecular markers like tumor HMGA2, MUC5AC/6, IDH1, PIWIL2, and DNA index exhibited prognostic significance. The review offers a comprehensive exploration of serum and tumor biomarkers in pCCA and their implications for prognosis. In my opinion, the authors covered almost all aspects, and the review at hand can serve as an easy guide to its readers and merits publication.

Author Response

We thank Reviewer 3 for their consideration and input on our manuscript. 

Reviewer 4 Report

Comments and Suggestions for Authors

The authors present a study aimed to identify biomarkers in blood, body fluids and tumour tissue that prognosticate overall survival and/or recurrence-free survival in patients who received curative intent surgery for pCCA.

The study is a well written summary of the most common biomarkers for pCCA. However, the manuscript can be improved by some changes.

The aim of the study “to identify biomarkers” could be changed to either “describe” or “summarize”, to better address the study concept. 

Could the authors please explain the reason for limiting the review to patients who receive curative intent surgery for pCCA as biomarkers ought to be as important in patients not receiving surgical treatment. 

Primary sclerosing cholangitis is an important risk factor for pCCA and ought to be mentioned in the manuscript. In addition, potential differences in biomarker profile depending on underlying liver disease could improve the manuscript. 

The manuscript could be further improved by addressing when biomarkers are most useful, optimal timing of measurements, the value of repeated measurements as well as the limitations of measuring biomarkers. 

Author Response

We thank Reviewer 4 for their thoughtful suggestions and have incorporated these into our manuscript as below.

Response to Comment 1 – We agree that the word "summarise" better addresses our study concept and thus have adjusted the manuscript accordingly (lines 21, 35, 84).

Response to comment 2- We agree that prognostic biomarkers are also important for non-curative pCCA patients. We wished to specifically address surgical patients, though, as we feel that incorporation of prognostically significant biomarkers into the preoperative staging of pCCA patients may improve patient selection for major liver resection. This would be particularly relevant for patients with resectable disease based on conventional preoperative staging, but borderline surgical fitness. We have highlighted this in the manuscript (lines 75-84). Our group ultimately aims to prospectively investigate and validate these biomarkers in patients undergoing pCCA resection.

Response to comment 3 - We have added a statement regarding the relevance of PSC in pCCA and association with evaluated biomarkers (lines 381-383). We also agree that differences in biomarker profile levels may exist according to patients’ comorbidities including liver disease, and this has been included in the discussion section of the manuscript (lines 476-479).

Response to comment 4 - We agree that time points for measuring biomarkers should be standardised. From a pragmatic perspective, these should be measured at diagnosis after biliary drainage. We do not feel there is great utility in measuring biomarkers post-operatively from a prognostic standpoint, but it may give an idea regarding tumour burden in patients with advanced disease. Dynamic trends of biomarker titres through serial measurements may provide further information regarding the aggressive of tumour biology, akin to the concept of “PSA velocity” in the setting of prostate cancer [1]. This has been added to our manuscript (lines 463-466, 471-476). In our manuscript, we have already highlighted biliary tract obstruction as a confounding factor in evaluating CA19-9 and bilirubin (lines 357-359, 203-204, 375-376). Ideally, such biomarkers should be measured in the non-jaundiced patient, after biliary drainage has been obtained. However, some units do not pursue routine pre-operative drainage. Moreover, awaiting normalisation of bilirubin may not be realistic and seen as an impediment to timely surgery in a patient with cancer (lines 376-381). Other biomarkers such as NLR and PLR may be confounded by concurrent inflammatory/infective conditions (e.g. cholangitis), and a recommendation for their measurement after resolution of such conditions has been added to our manuscript (line 411).

Reference

1. D'Amico AV, Chen MH, Roehl KA, Catalona WJ. Preoperative PSA velocity and the risk of death from prostate cancer after radical prostatectomy. N Engl J Med. 2004;351(2):125-35.